# Pulmonary Pharmacokinetic and Pharmacodynamic Evaluation of Ampicillin/Sulbactam Regimens for Pneumonia Caused by Various Bacteria, including *Acinetobacter baumannii*

**DOI:** 10.3390/antibiotics12020303

**Published:** 2023-02-02

**Authors:** Tetsushu Onita, Kazuro Ikawa, Noriyuki Ishihara, Hiroki Tamaki, Takahisa Yano, Kohji Naora, Norifumi Morikawa

**Affiliations:** 1Department of Clinical Pharmacotherapy, Hiroshima University, 1-2-3 Kasumi, Minami-ku, Hiroshima 734-8551, Japan; 2Department of Pharmacy, Shimane University Hospital, 89-1 Enya, Izumo 693-8501, Japan

**Keywords:** ampicillin, sulbactam, pharmacokinetics, pharmacodynamics, pulmonary

## Abstract

This study aimed to assess the dosing regimens of ampicillin/sulbactam for pneumonia based on pulmonary pharmacokinetic (PK)/pharmacodynamic (PD) target attainment. Using the literature data, we developed pulmonary PK models and estimated the probabilities of attaining PK/PD targets in lung tissue. Against bacteria other than *A. baumannii* (the general treatment), the PK/PD target was set as both 50% time above the minimum inhibitory concentration (T > MIC) for ampicillin and 50% T > 0.5 MIC for sulbactam. For the *A. baumannii* treatment, the PK/PD target was set as 60% T > MIC for sulbactam. The pulmonary PK/PD breakpoint was defined as the highest minimum inhibitory concentration (MIC) at which the target attainment probability in the lung tissue was ≥90%. The lung tissue/serum area under the drug concentration–time curve from 0 to 3 h (AUC_0–3h_) ratios for ampicillin and sulbactam were 0.881 and 0.368, respectively. The ampicillin/sulbactam AUC_0–3h_ ratio in the lung tissue was 3.89. For the general treatment, the pulmonary PK/PD breakpoint for ampicillin/sulbactam at 3 g four times daily in typical patients with creatinine clearance (CL_cr_) of 60 mL/min was 2 μg/mL, which covered the MIC_90s_ (the MICs that inhibited the growth of 90% of the strains) of most gram-positive and gram-negative bacteria. For the *A. baumannii* treatment, the pulmonary PK/PD breakpoint for ampicillin/sulbactam at 9 g 4-h infusion three times daily (27 g/day) in patients with a CL_cr_ of 60 mL/min was 4 μg/mL, which covered the MIC_90_ of *A. baumannii*. A PK/PD evaluation for pneumonia should be performed in the lung tissue (the target site) rather than in the blood because sulbactam concentrations are lower in lung tissue. These findings should facilitate the selection of ampicillin/sulbactam regimens for pneumonia caused by various bacteria, including *A. baumannii*.

## 1. Introduction

Ampicillin/sulbactam, containing the β-lactam antimicrobial agent, ampicillin, and the β-lactamase inhibitor, sulbactam, has been widely used at a dose ratio of 2:1. Both ampicillin and sulbactam are water-soluble drugs and mainly excreted by the kidneys [1]. Ampicillin/sulbactam has been used as a first-line treatment for pneumonia and preoperative prophylaxis for pneumonectomy [1,2,3,4]. 

Bacterial pneumonia, which is caused by a wide variety of bacteria and is broadly classified into community-acquired, hospital-acquired, aspiration pneumonia, etc., is one of the most common infections globally. Community-acquired pneumonia is caused by bacteria such as *Streptococcus pneumoniae*, *Haemophilus influenzae*, and *Moraxella catarrhalis*. Ampicillin/sulbactam is frequently prescribed for community-acquired pneumonia because of its antibacterial activity against β-lactamase-producing pathogens [5]. Furthermore, ampicillin/sulbactam has been used to treat aspiration pneumonia mainly caused by oral bacteria such as the *Streptococcus anginosus* group, *Peptostreptococcus* species, *Prevotella* species, and *Fusobacterium* species [6,7]. Meanwhile, *Acinetobacter baumannii* is a globally important pathogen that causes infectious diseases such as ventilator-related pneumonia [8,9]. Although ampicillin/sulbactam is prescribed to treat *A. baumannii* infection, the strain is resistant to ampicillin, but sulbactam is known to exhibit antibacterial activity against *A. baumannii*, independently of ampicillin [10,11]. Therefore, ampicillin/sulbactam has been widely used for the treatment of pneumonia caused by various bacteria. 

However, ampicillin/sulbactam at 12 g/day, as compared with 6 g/day, increased the incidence of hepatobiliary enzyme elevation [12]. Therefore, dose-dependent side effects cannot be excluded, and dosage adjustment considering each patient’s characteristics (e.g., renal function) should be required. Furthermore, according to the pharmacokinetic (PK)/pharmacodynamic (PD) theory, the exposure time during which the drug concentrations remain above the minimum inhibitory concentration (MIC) for bacteria (T > MIC) is correlated with the antibacterial activity of β-lactams such as ampicillin [13,14]. The efficacy should be improved with appropriately adjusted regimens, according to the MIC for the bacteria.

In blood, optimization of the ampicillin/sulbactam dosage using population PK/PD analysis has been reported [15]. However, for pneumonia, antimicrobial agents act in the lung tissues, such as the alveoli, in which the main causative pathogens are present, rather than in the blood. Therefore, dose optimization using pulmonary PK/PD analysis might be useful. Several previous reports described the penetration of ampicillin and sulbactam into lung tissue [16] and the alveolar lining fluid [17]. However, they did not fully describe PK in the lungs, and no accurate pulmonary PK/PD evaluation using a mathematical model has been performed. 

Therefore, this study characterized pulmonary PK models of ampicillin and sulbactam using previously reported data, such as PK and physiological parameters, and evaluated ampicillin/sulbactam pulmonary PK/PD target attainment considering both ampicillin and sulbactam concentrations in lung tissue. Moreover, we optimized dosing regimens of ampicillin/sulbactam for pneumonia caused by various bacteria, including *A. baumannii*.

## 2. Results

### 2.1. Lung Tissue/Serum Ratio and Ampicillin/Sulbactam Ratio in Serum and Lung Tissue

The lung tissue/serum ratio and ampicillin/sulbactam ratio in the serum and lung tissue are presented in Table 1.

The C_max_ values for ampicillin and sulbactam from the literature were 40.8 and 25.3 μg/mL in the serum and 35.6 and 8.6 μg/g in the lung tissue, respectively. The area under the drug concentration–time curve from the 0 to 3 h (AUC_0–3h_) values for ampicillin and sulbactam, calculated from the literature, were 83.5 and 51.2 μg·h/mL in the serum and 73.6 and 18.9 μg·h/g in the lung tissue, respectively. For ampicillin, the lung tissue/serum ratios were 0.873 for the C_max_ and 0.881 for the AUC_0–3h_. For sulbactam, the lung tissue/serum ratios were 0.339 for the C_max_ and 0.368 for the AUC_0–3h_. For the pulmonary PK modeling, the KP_lung_ for ampicillin/sulbactam was fixed as 0.881/0.368 for the AUC_0–3h_. The ampicillin/sulbactam ratio of the C_max_ and AUC_0–3h_ in the lung tissue was approximately 4.

### 2.2. Model Validation

Visual predictive checks were also performed for the observed and predicted lung tissue concentration vs. the time curves of ampicillin and sulbactam (Figure 1). The mean ± standard deviation of the observed lung tissue concentrations was almost within the predicted 95% confidence intervals for the 2.5th, 50th, and 97.5th percentiles.

### 2.3. PK and PD Evaluation

The probabilities of target attainment in the lung tissue using different ampicillin/sulbactam regimens at specific MICs are presented in Figure 2 for bacteria other than *A. baumannii* (the general treatment). Regarding the probability–MIC curve in the lung tissue for ampicillin, only the MICs with a target attainment probability of more than 90% for sulbactam (50% T > 0.5 MIC for sulbactam) are represented by red symbols, indicating that the ampicillin/sulbactam combination is effective. The results of Figure 2 demonstrate that the number of effective regimens for the ampicillin/sulbactam combination decreases with an improving renal function or as the MIC of the causative pathogen increases. Furthermore, the pulmonary PK/PD breakpoints (the highest MIC at which the target attainment probability in the lung tissue was ≥90%) are presented in Table 2. The pulmonary PK/PD breakpoints of 3.0 g four times daily were 1 μg/mL for CL_cr_ = 90 mL/min, 2 μg/mL for CL_cr_ = 60 mL/min, and 8 μg/mL for CL_cr_ = 30 mL/min, respectively.

Moreover, the probabilities of target attainment in the lung tissue are presented in Figure 3 for the *A. baumannii* treatment with sulbactam activity. The pulmonary PK/PD breakpoints are presented in Table 3. The pulmonary PK/PD breakpoints of 9.0 g three times daily with 4-h infusion in the lung tissue were 2 μg/mL for CL_cr_ = 90 mL/min, 4 μg/mL for CL_cr_ = 60 mL/min, and 16 μg/mL for CL_cr_ = 30 mL/min, respectively.

## 3. Discussion

No pulmonary PK/PD evaluation with a mathematical method has been reported for ampicillin/sulbactam. This study developed site-specific PK models, which are important for pneumonia.

For both the C_max_ and AUC, the lung tissue/plasma ratios of sulbactam were < half that for ampicillin. From these results, the penetration of sulbactam from the systemic circulation into the lung tissue was lower than that of ampicillin. The ampicillin/sulbactam ratio resulting in the best antibacterial activity ranged from 1.0 to 2.0 [18]. Therefore, for general treatment, if the PK/PD target in the lung tissue is attained for ampicillin but not for sulbactam, the combination treatment (ampicillin/sulbactam) will not be effective. Thus, for general treatment, it is necessary to analyze the pulmonary pharmacokinetics/pharmacodynamics of both drugs for the prediction of the efficacy of the combination regimen. 

Pulmonary PK models for ampicillin and sulbactam were described using hybrid modeling, which can incorporate previously reported PK and physiological parameters. Since a specific organ clearance depends mainly on its physiological organ blood flow, both the system-to-lung clearance and lung-to-system clearance were assumed to be the same and set as the lung blood flow in the hybrid modeling. Independently from the conventional PK model, the mass balance in the lung compartment was assumed not to affect the mass balance in the central and peripheral compartments. The visual predictive check plots (Figure 1) indicated that the most observed lung concentrations were within model-predicted ranges. However, the observed lung concentrations currently available (mean ± standard deviation at three-time points) were too few to assess the model performance. Furthermore, model validation by goodness-of-fitness plots was not possible due to a lack of individual raw concentration data in the literature [16]. The model in this study, thus, needs to be further validated in the future.

Next, the PK/PD target attainment in the lung tissue for different dosing regimens was estimated. For general treatment, the probability of target attainment for ampicillin in the lung tissue (the pulmonary PK/PD breakpoint of 3 g four times daily was 2 μg/mL for CL_cr_ = 60 mL/min, Table 2) was in line with the previously published data in the plasma (the PK/PD breakpoint of 3 g four times daily in plasma was 2 μg/mL for CL_cr_ = 60 mL/min, [19]). However, because the sulbactam concentration in the lung tissue and the probability of target attainment for sulbactam are lower, some regimens are considered to be poorly active as combination treatments, even though ampicillin had a probability of target attainment exceeding 90% (Figure 2). Furthermore, the probability of target attainment for sulbactam in the lung tissue (the pulmonary PK/PD breakpoint of 6 g four times daily was 1 μg/mL for CL_cr_ = 60 mL/min, Table 3) was lower than that reported in the plasma (the PK/PD breakpoint of 6 g four times daily in plasma was 4 μg/mL for CL_cr_ = 60 mL/min [20]). Therefore, considering the pharmacokinetics/pharmacodynamics of ampicillin/sulbactam in lung tissue, the efficacy of this combination might depend on the probability of target attainment for sulbactam rather than ampicillin. 

Regarding the general treatment, the pulmonary PK/PD breakpoints of ampicillin/sulbactam 3 g four times daily (12 g/day: the approved maximum dosage) were 1 μg/mL (the MIC_90_ [the MIC that inhibited the growth of 90% of the strains] of the MSSA) for CL_cr_ = 90 mL/min, 2 μg/mL (the MIC_90_ of the *S. pneumoniae* and *Prevotella* species) for CL_cr_ = 60 mL/min, and 8 μg/mL for CL_cr_ = 30 mL/min. Thus, the probability of attaining the pulmonary PK/PD target decreased as renal function improved. For CL_cr_ = 30 mL/min, the PK/PD breakpoints of the twice daily dosing regimen were lower in the lung tissue (0.5 μg/mL for 1.5 g twice daily and 1 μg/mL for 3.0 g twice daily) than in the plasma (1 μg/mL for 1.5 g twice daily and 2 μg/mL for 3.0 g twice daily) [19] because of the poor penetration of sulbactam into the lung tissue. Suzuki et al. reported that ampicillin/sulbactam treatment for elderly patients with pneumonia and renal dysfunction (10 mL/min ≤ CL_cr_ < 50 mL/min) was more effective using a four-times daily regimen than using a twice-daily regimen [21]. Similarly, from our findings that the pulmonary PK/PD breakpoints of the four times daily regimen were higher than those of the twice-daily regimen, we recommend four times daily regimens for patients with renal dysfunction; 3 g four times daily (12 g/day) as the maximum dose for pneumonia is recommended by the Japanese Association for Infectious Diseases/Japanese Society of Chemotherapy guidelines for the clinical management of infectious diseases [6]. In this study, the pulmonary PK/PD breakpoint of 3 g four times daily was 2 μg/mL for CL_cr_ = 60 mL/min, and it covered the MIC_90s_ of the MSSA, *S. pneumoniae*, *M. catarrhalis*, the *S. anginosus* group, the *Peptostreptococcus* species, the *Prevotella* species, and the *Fusobacterium* species. Therefore, this guideline dose might be valid as empiric therapy for community-acquired and aspiration pneumonia. However, because the MIC for β-lactamase–nonproducing ampicillin-resistant *H. influenzae* is high (MIC_90_ = 8 μg/mL), the use of other antimicrobial agents might be required for typical patients with normal renal function (CL_cr_ ≥ 60 mL/min).

Regarding the *A. baumannii* treatment, the pulmonary PK/PD breakpoints of ampicillin/sulbactam 3 g four times daily (12 g/day: the approved maximum dosage) were 0.25 μg/mL for CL_cr_ = 90 mL/min, 0.5 μg/mL for CL_cr_ = 60 mL/min, and 2 μg/mL for CL_cr_ = 30 mL/min. From these results, the maximum approved dose does not appear to achieve 4 μg/mL (the MIC_90_ of *A. baumannii*) because of the low penetration of sulbactam into the lung tissue. This suggests that dosing regimens for ampicillin/sulbactam exceeding the maximum approved dose are required. A clinical report also found that a high-dose regimen was effective in patients with ventilator-associated pneumonia caused by multidrug-resistant *A. baumannii*. [22]. Furthermore, the Sanford guideline recommends a 4-h infusion of ampicillin/sulbactam at 9 g three times daily (27 g/day) for ventilator-related pneumonia caused by *A. baumannii* [23]. Our results illustrated that this dosing regimen achieved an MIC_90_ of 4 μg/mL against *A. baumannii* in typical patients with CL_cr_ = 60 mL/min. Thus, from the perspective of the pulmonary penetration of sulbactam, it was confirmed that high-dose regimens are necessary for the treatment of *A. baumannii*. 

Concerning the limitations of this study, the PK and physiological parameters derived from previous reports [16,19,24,25] were estimated from uninfected patients or healthy volunteers. The inflammation of lung tissue caused by pneumonia can increase vascular permeability in the lungs. Therefore, the population parameters used in this study might underestimate the pulmonary penetration of ampicillin and sulbactam. Second, because our findings are only predictions of efficacy based on PK/PD simulations, it is necessary to perform clinical studies in infected patients to identify the appropriateness of the dosing regimens. The hybrid model used in our study enables the analysis of pharmacokinetics/pharmacodynamics at various target sites using the literature data, such as the organ blood flow, organ volume, and tissue/blood drug concentration ratios. This method is practical and versatile in clinical situations involving difficult tissue sampling.

## 4. Materials and Methods

### 4.1. Pulmonary PK Modeling for Ampicillin and Sulbactam

The pulmonary pharmacokinetics of ampicillin and sulbactam were separately described for each drug using the following hybrid model (Figure 4). The hybrid model is a model in which physiological parameters such as organ blood flow and volume are partially connected to the conventional PK model. Based on blood concentrations, this model has been used for the target site PK/PD analysis [19,26,27,28,29]. In this study, a lung compartment was connected to a two-compartment PK model using blood concentrations [19]; thus, the hybrid model consisted of three compartments.
dX(central)/dt = R_inf_ − (CL/V_central_ + Q/V_central_) × X(central) + Q × X(peripheral)/V_peripheral_
dX(peripheral)/dt = Q × X(central)/V_central_ − Q × X(peripheral)/V_peripheral_
dX(lung)/dt = Q_lung_ × X(central)/V_central_ − Q_lung_ × X(lung)/V_lung_/KP_lung_

In the formulas, X(central), X(peripheral), and X(lung) are the amounts of the drug (mg) in the central, peripheral, and lung compartments, respectively; R_inf_ is the rate of infusion (mg/h); CL is the clearance (L/h) from the central compartment; V_central_ and V_peripheral_ are the volumes of distribution (L) of the central and peripheral compartments, respectively; and Q is the central–peripheral intercompartmental clearance (L/h). The model parameters of ampicillin and sulbactam are listed in Table 4. In the analysis of blood concentrations, the fixed-effects parameters (*θ*CL, *θ*V_central_, *θ*Q, and *θ*V_peripheral_) and the interindividual variability (ηCL, ηV_central_, ηQ, and ηV_peripheral_) were derived from previously reported population PK parameters of ampicillin and sulbactam [19]. The fixed-effects parameters, *θ*i (CL, V_central_, Q, and V_peripheral_), of ampicillin and sulbactam in blood concentrations, were fixed as follows: CL = 11.03 and 10.50 (L/h), V_central_ = 7.80 and 8.96 (L), Q = 7.07 and 7.29 (L/h), and V_peripheral_ = 3.98 and 4.93 (L). In addition, CL_cr_, which was calculated by the Cockcroft–Gault formula, was incorporated as a covariate of the CL. We also used Q_lung_ (lung plasma flow in L/h) and V_lung_ (lung volume in kg) as physiological parameters. Q_lung_ and V_lung_ were derived from reference values [24,25]. For both drugs, the physiological fixed-effects parameters were fixed as follows: Q_lung_ = 207 (L/h), and V_lung_ = 0.47 (L). Since the drug is present only in the plasma portion, lung plasma flow was calculated by multiplying the lung blood flow (360 L/h) [24,25] and human hematocrit value (approximately 42.5%), as follows: Q_lung_ = lung blood flow (360 L/h) ∗ (1–0.425) = 207 (L/h). The result of the lung tissue/serum AUC_0–3h_ ratio calculated in the results section was used as the KP_lung_ (lung-to-plasma partition coefficient). The demographic information of the literature data used in this study is represented in the Appendix A. PK modeling predicting the drug concentrations in lung tissue was performed using the NONMEM program (version 7.4; ICON Public Limited Company, Dublin, Ireland).

### 4.2. Calculation of the Lung Tissue/Serum Ratio and Ampicillin/Sulbactam Ratio in Serum and Lung Tissue

The mean values of lung tissue and serum concentrations (1, 1.5, and 2–4 h each) in patients reported by Frank et al. [16] were used because individual raw data were not described. For each drug, the C_max_ was defined as the highest value in the mean concentrations described in the literature. The AUC_0–3h_, based on the mean concentrations (three-time points), was estimated according to the trapezoidal rule. The AUC_0–inf_ was not estimated because of insufficient time points for appropriate extrapolation to infinity. The lung tissue/serum ratio and ampicillin/sulbactam ratio in the serum and lung tissue were calculated from the C_max_ or AUC_0–3h_ ratio. The specific gravity of lung tissue was assumed to be 1 (g = mL).

### 4.3. Model Validation

Visual predictive checks were performed to validate the models. One thousand datasets were simulated using the model parameters, including the interindividual and residual variabilities. The observed mean ± standard deviation values in the previous report [16] were confirmed whether they were within the 95% confidential interval of the predicted concentration in lung tissue.

### 4.4. PK/PD Simulation

A set of fixed-effects parameters, *θ*i (CL, V_central_, Q, V_peripheral_, KP_lung_, Q_lung_, and V_lung_), of 1000 virtual subjects for each renal function (three typical CL_cr_ = 90, 60, and 30 mL/min) were randomly generated using the $SIMULATION command in NONMEM based on each mean value and interindividual variability. By describing each dosing information in the dataset, ampicillin and sulbactam concentrations in lung tissue were calculated by each dosing regimen. The time at which the drug concentration coincided with a specific MIC (0.06–64 μg/mL) was determined, and T > MIC was calculated as the cumulative percentage of time over a 24-h period for different renal functions and different dosing intervals in lung tissue. The total concentration was not able to be corrected for the free fraction because the tissue protein-binding of both drugs in lung tissue is currently unknown. For general treatment, the probability of target attainment (%) at a specific MIC in lung tissue was defined as the proportion of 1000 estimates that achieved the bactericidal target (both 50% T > MIC for ampicillin [30,31] and 50% T > 0.5 MIC for sulbactam). This target was chosen because the MIC evaluation in vitro for ampicillin/sulbactam is 2:1. [32] For the *A. baumannii* treatment, the probability of target attainment (%) at a specific MIC in lung tissue was defined as the proportion of 1000 estimates that achieved the bactericidal target (60% T > MIC for sulbactam) [33].

The MIC distribution data for ampicillin/sulbactam were obtained from the Japanese surveillance of antimicrobial susceptibilities [20,34,35,36]. Nine common types of pathogens were selected for pneumonia. These included methicillin-susceptible *Staphylococcus aureus* (MSSA; n = 676, the MIC for the 90th percentile of the clinical strains [MIC_90_] = 1 μg/mL), *S. pneumoniae* (n = 565, MIC_90_ = 2 μg/mL), *H. influenzae* (all strains: n = 544, MIC_90_ = 4 μg/mL; β-lactamase-nonproducing ampicillin-resistant *H. influenzae*: n = 70, MIC_90_ = 8 μg/mL), *M. catarrhalis* (n = 491, MIC_90_ = 0.25 μg/mL), the *S. anginosus* group (n = 100, MIC_90_ = 0.25 μg/mL), the *Peptostreptococcus* species (n = 100, MIC_90_ = 0.03 μg/mL), the *Prevotella* species (n = 100, MIC_90_ = 2 μg/mL), the *Fusobacterium* species (n = 50, MIC_90_ = 0.06 μg/mL) for ampicillin/sulbactam, and *A. baumannii* (n = 27, MIC_90_ = 4 μg/mL) for sulbactam.

## 5. Conclusions

This study focused on lung tissue as the target site of pneumonia, and we performed a pulmonary PK/PD evaluation. For pneumonia, pharmacokinetics/pharmacodynamics should be evaluated in the lungs rather than in the blood because the sulbactam concentration in lung tissue is low. We provided practical ampicillin/sulbactam dosing regimens for pneumonia caused by various pathogens, considering the susceptibility of pathogens and renal function.

## Figures and Tables

**Figure 1 antibiotics-12-00303-f001:**
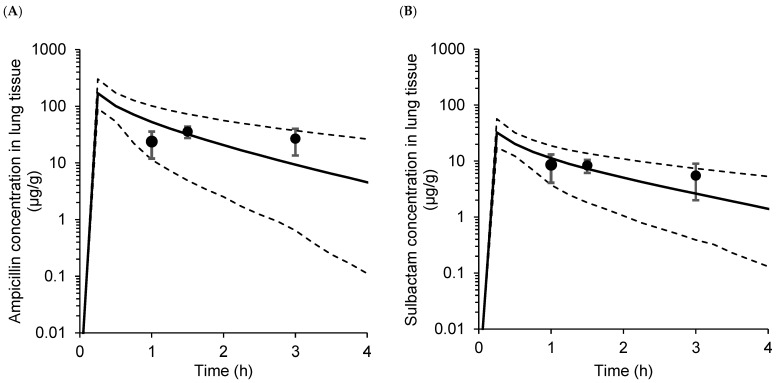
Visual predictive check plots of (**A**) ampicillin and (**B**) sulbactam, representing the observed lung tissue concentrations (mean ± standard deviation) after a 15-min of infusion of ampicillin/sulbactam 3 g (ampicillin 2 g and sulbactam 1 g) derived from the literature data [16]. The heavy line and dotted line denote the median and the 95% predicted interval, calculated from 1000 replicates, respectively.

**Figure 2 antibiotics-12-00303-f002:**
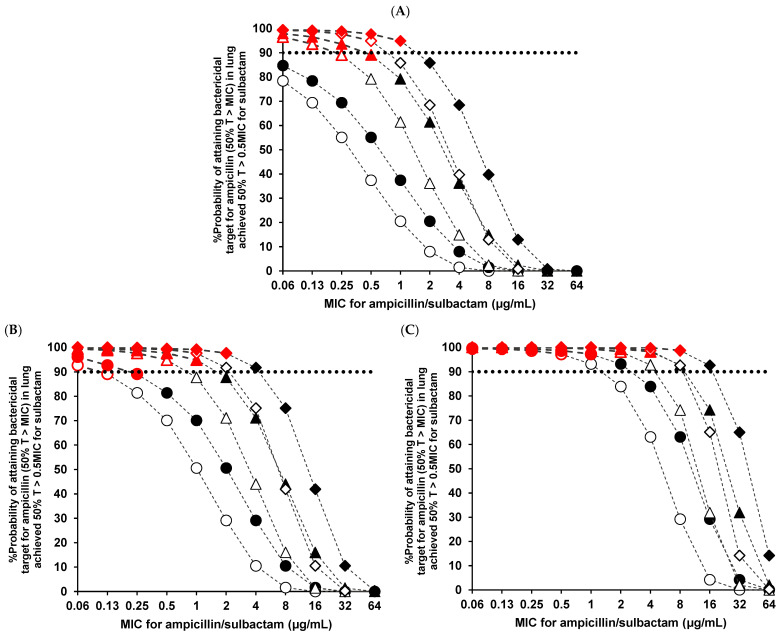
Probabilities of attaining bactericidal (50% T > MIC for ampicillin) targets in lung tissue at specific MICs for typical patients of (**A**) CL_cr_ = 90 mL/min, (**B**) CL_cr_ = 60 mL/min, and (**C**) CL_cr_ = 30 mL/min, respectively. Regarding the probability–MIC curve for ampicillin in lung tissue, MICs with a probability of target attainment for sulbactam of more than 90% (50% T > 0.5 MIC for sulbactam), indicating that the ampicillin/sulbactam combination is effective, are represented by red symbols. The lines represent the probability of target attainment for each dosing regimen of ampicillin/sulbactam (〇 1.5 g twice daily, 0.5-h infusion; ● 3.0 g twice daily, 0.5-h infusion; △ 1.5 g three times daily, 0.5-h infusion; ▲ 3.0 g three times daily, 0.5-h infusion; ◇ 1.5 g four times daily, 0.5-h infusion; ◆ 3.0 g four times daily, 0.5-h infusion). The dotted black line represents a 90% probability. MIC, minimum inhibitory concentration.

**Figure 3 antibiotics-12-00303-f003:**
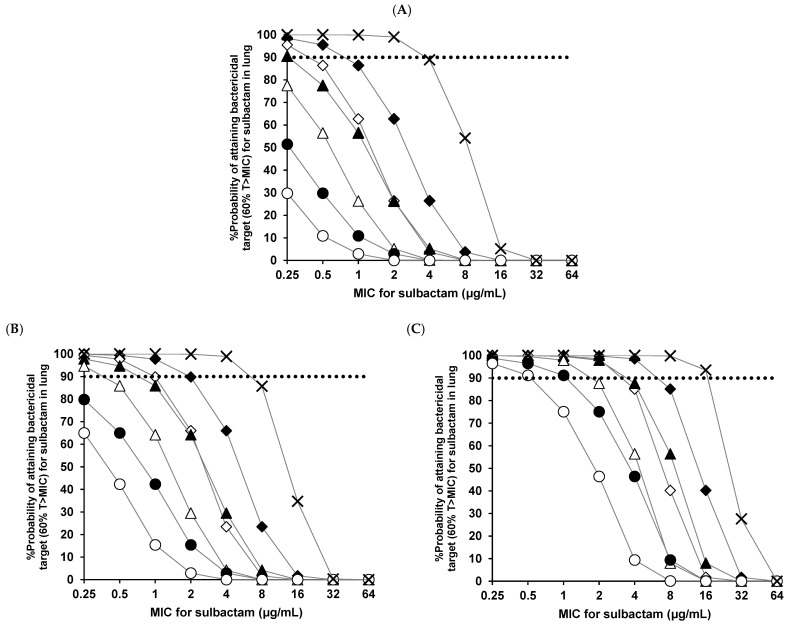
Probabilities of attaining bactericidal (60% T > MIC for sulbactam) targets in lung tissue at specific MICs for typical patients of (**A**) CL_cr_ = 90 mL/min, (**B**) CL_cr_ = 60 mL/min, and (**C**) CL_cr_ = 30 mL/min, respectively. The lines represent the probability of target attainment for each dosing regimen of ampicillin/sulbactam (〇 3.0 g twice daily, 0.5-h infusion; ● 6.0 g twice daily, 0.5-h infusion; △ 3.0 g three times daily, 0.5-h infusion; ▲ 6.0 g three times daily, 0.5-h infusion; ◇ 3.0 g four times daily, 0.5-h infusion; ◆ 6.0 g four times daily, 0.5-h infusion; × 9.0 g three times daily, 4-h infusion). The dotted black line represents a 90% probability. MIC, minimum inhibitory concentration.

**Figure 4 antibiotics-12-00303-f004:**
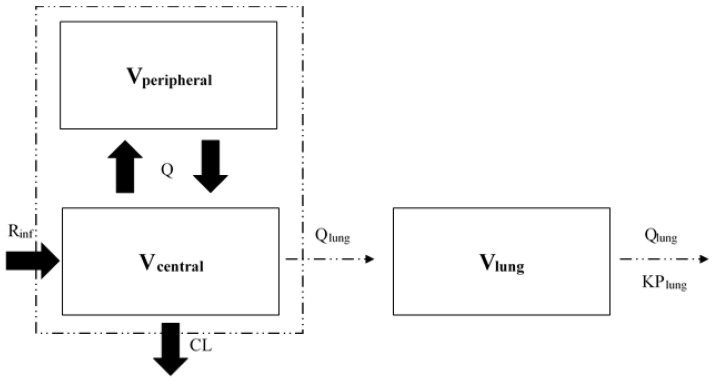
Pulmonary PK modeling of ampicillin and sulbactam.Model parameters: V_central_ and V_peripheral_, volumes of distribution of the central and peripheral compartments (L), respectively; V_lung_, lung volume (kg); CL, clearance (L/h); Q, central–peripheral intercompartmental clearance (L/h); Q_lung_, lung plasma flow (L/h); KP_lung_, lung-to-plasma partition coefficient; R_inf_, rate of infusion (mg/h).

**Table 1 antibiotics-12-00303-t001:** Calculation of lung tissue/serum ratio and ampicillin/sulbactam ratio in the lung tissue.

Specimen and Parameter	Value		
	Ampicillin 2.0 g (15 subjects)	Sulbactam 1.0 g (15 subjects)	Ampicillin/ Sulbactam Ratio (2.0 g/1.0 g)
Serum			
C_max_ (μg/mL)	40.8	25.3	1.61
AUC_0–3h_ (μg·h/mL)	83.5	51.2	1.63
Lung tissue			
C_max_ (μg/g)	35.6	8.6	4.14
AUC_0–3h_ (μg·h/g)	73.6	18.9	3.89
Lung tissue/serum ratio			
C_max_	0.873	0.339	
AUC_0–3h_	0.881	0.368	

AUC_0–3h_, area under the drug concentration–time curve from 0 to 3 h, calculated based on the trapezoidal rule; C_max_, observed maximum concentration. Data are provided as the mean derived from [16].

**Table 2 antibiotics-12-00303-t002:** Pulmonary PK/PD breakpoints of ampicillin/sulbactam for general treatment with both activities of bactericidal ampicillin and β-lactamase-inhibiting sulbactam.

Ampicillin/Sulbactam Regimen	Bactericidal Target (50% T > MIC for Ampicillin) and 50% T > 0.5 MIC for Sulbactam
CL_cr_ = 90 mL/min	
1.5 g twice daily, 0.5-h infusion (total 3 g/day)	-
3.0 g twice daily, 0.5-h infusion (total 6 g/day)	-
1.5 g three times daily, 0.5-h infusion (total 4.5 g/day)	0.13
3.0 g three times daily, 0.5-h infusion (total 9 g/day)	0.25
1.5 g four times daily, 0.5-h infusion (total 6 g/day)	0.5
3.0 g four times daily, 0.5-h infusion (total 12 g/day)	1
CL_cr_ = 60 mL/min	
1.5 g twice daily, 0.5-h infusion (total 3 g/day)	0.06
3.0 g twice daily, 0.5-h infusion (total 6 g/day)	0.13
1.5 g three times daily, 0.5-h infusion (total 4.5 g/day)	0.5
3.0 g three times daily, 0.5-h infusion (total 9 g/day)	1
1.5 g four times daily, 0.5-h infusion (total 6 g/day)	1
3.0 g four times daily, 0.5-h infusion (total 12 g/day)	2
CL_cr_ = 30 mL/min	
1.5 g twice daily, 0.5-h infusion (total 3 g/day)	0.5
3.0 g twice daily, 0.5-h infusion (total 6 g/day)	1
1.5 g three times daily, 0.5-h infusion (total 4.5 g/day)	2
3.0 g three times daily, 0.5-h infusion (total 9 g/day)	4
1.5 g four times daily, 0.5-h infusion (total 6 g/day)	4
3.0 g four times daily, 0.5-h infusion (total 12 g/day)	8

Note: Pulmonary PK/PD breakpoints are defined as the highest MIC attaining more than 90% probabilities in lung tissue.

**Table 3 antibiotics-12-00303-t003:** Pulmonary PK/PD breakpoints of ampicillin/sulbactam for *A. baumannii* treatment with sulbactam activity.

Ampicillin/Sulbactam Regimen	Bactericidal Target 60% T > MIC for Sulbactam
CL_cr_ = 90 mL/min	
3.0 g twice daily, 0.5-h infusion (total 6 g/day)	-
6.0 g twice daily, 0.5-h infusion (total 12 g/day)	-
3.0 g three times daily, 0.5-h infusion (total 9 g/day)	0.13
6.0 g three times daily, 0.5-h infusion (total 18 g/day)	0.25
3.0 g four times daily, 0.5-h infusion (total 12 g/day)	0.25
6.0 g four times daily, 0.5-h infusion (total 24 g/day)	0.5
9.0 g three times daily, 4-h infusion (total 27 g/day)	2
CL_cr_ = 60 mL/min	
3.0 g twice daily, 0.5-h infusion (total 6 g/day)	-
6.0 g twice daily, 0.5-h infusion (total 12 g/day)	0.06
3.0 g three times daily, 0.5-h infusion (total 9 g/day)	0.25
6.0 g three times daily, 0.5-h infusion (total 18 g/day)	0.5
3.0 g four times daily, 0.5-h infusion (total 12 g/day)	0.5
6.0 g four times daily, 0.5-h infusion (total 24 g/day)	1
9.0 g three times daily, 4-h infusion (total 27 g/day)	4
CL_cr_ = 30 mL/min	
3.0 g twice daily, 0.5-h infusion (total 6 g/day)	0.5
6.0 g twice daily, 0.5-h infusion (total 12 g/day)	1
3.0 g three times daily, 0.5-h infusion (total 9 g/day)	1
6.0 g three times daily, 0.5-h infusion (total 18 g/day)	2
3.0 g four times daily, 0.5-h infusion (total 12 g/day)	2
6.0 g four times daily, 0.5-h infusion (total 24 g/day)	4
9.0 g three times daily, 4-h infusion (total 27 g/day)	16

Note: Pulmonary PK/PD breakpoints are defined as the highest MIC attaining more than 90% probabilities in lung tissue.

**Table 4 antibiotics-12-00303-t004:** PK parameters predicting lung tissue concentration.

Parameter	Ampicillin	Sulbactam
	Value (RSE%)	Value (RSE%)
Fixed-effects parameter		
CL (L/h) = *θ*_CL_ × (CL_cr_/68.3)*^θ^*^CLcr on CL^		
*θ*_CL_(L/h) ^a^	11.03 (5.1)	10.50 (5.0)
*θ*_CLcr on CL_ ^a^	0.831 (14.1)	0.774 (18.6)
V_central_ (L) = *θ*_Vcentral_ ^a^	7.80 (5.9)	8.96 (9.6)
Q (L/h) = *θ*_Q_ ^a^	7.07 (14.3)	7.29 (21.4)
V_peripheral_ (L) = *θ*_Vperipheral_ ^a^	3.98 (12.3)	4.93 (13.4)
KP_lung_ = *θ*_KPlung_ ^b^	0.881 Fixed	0.368 Fixed
Q_lung_(L/h) = *θ*_Qlung_ ^c^	207 Fixed	207 Fixed
V_lung_(kg) = *θ*_Vlung_ ^c^	0.47 Fixed	0.47 Fixed
Interindividual variability (exponential error model)		
*ηCL* ^a^	0.0985 (26.1)	0.0626 (26.8)
*ηVcentral* ^a^	0.160 (21.3)	0.147 (27.5)
*ηQ* ^a^	0.588 (44.2)	0.399 (48.4)
*ηVperipheral* ^a^	0.298 (37.2)	0.177 (37.9)
Residual variability (additive error model)		
ε ^a^	2.70 (26.2)	1.22 (38.8)

^a^, Parameters derived from [19]; ^b^, parameter derived from the lung tissue/serum AUC_0–3h_ ratio (Table 1); ^c^, parameter derived from [24,25]. RSE, relative standard error; *θ*, population mean value; η, random variable, which is normally distributed with a mean of zero and variance; ε, random error, which is normally distributed with a mean of zero and variance. CL_cr_ was calculated by the Cockcroft–Gault formula.

## Data Availability

Data supporting the findings of this study were derived from the resource available in the public domain.

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
