# Peer review of "Pulmonary Pharmacokinetic and Pharmacodynamic Evaluation of Ampicillin/Sulbactam Regimens for Pneumonia Caused by Various Bacteria, including Acinetobacter baumannii"

_antibiotics, 2023, doi:10.3390/antibiotics12020303_

Round 1

Reviewer 1 Report

In this study, an ampicillin/sulbactam PKPD model was constructed using NONMEM to determine whether various dosing regimens effectively treat pneumonia. The efficacy assessment index was mainly time over MIC (T > MIC) as the drugs are antibiotics. Parameters and data used in constructing the model used existing literature values. This article is meaningful because it was intended to predict and evaluate drug concentration at the target site. I have several comments, however:

1. The readability of the Discussion section is too poor. Paragraph breaks are necessary. The story flow should be cut off appropriately. There are overlapping contents also. A significant revision to improve simplicity is required.

2. Why did authors see AUC0-3hour? Why not compare AUClast or AUCinf?

3. Please add Goodness-of-Fif plots, not only VPCs. There are only three prediction points in the VPCs. Why only three points?

4. Please also add PK for ampicillin and sulbactam, respectively in the Introduction section.

5. What formula was used for CLcr? CG or MDRD or other?

6. Abbreviations should be used consistently. Use of the "PKPD" abbreviation is intermittent.

7. Why is this model a hybrid? How many compartments are there in this model?

8. Table 4: Fix effects parameter (X) -> Fixed ~. Several semicolons are missing in the footnote.

9. Various dosing regimen simulations were attempted. Ampicillin/sulbactam is already widely used clinically. What does it mean to subdivide the regimen by MIC and CLcr?

Author Response

Thank you for your comments. The text shown in red below is our response to your thought-provoking comments. Line (L) numbers are shown as those in the revised manuscript. We would be grateful if you could re-review it. Please also refer to the attached file.

Comment 1. The readability of the Discussion section is too poor. Paragraph breaks are necessary. The story flow should be cut off appropriately. There are overlapping contents also. A significant revision to improve simplicity is required.

Response 1.

I fixed the Discussion section to read the paper easily. The paragraph breaks were created in each story. Also, overlapping contents were deleted to simplify the manuscript. Please review the Discussion section.

Comment 2. Why did authors see AUC0-3hour? Why not compare AUClast or AUCinf?

Response 2.

I chose AUC0-3h because of insufficient timepoints for appropriate extrapolation to infinity. I added that in the manuscript.

L392 (Materials and Methods)

AUC0–inf was not estimated because of insufficient timepoints for appropriate extrapolation to infinity.

Comment 3. Please add Goodness-of-Fif plots, not only VPCs. There are only three prediction points in the VPCs. Why only three points?

Response 3.

I was not able to perform Goodness-of-fitness plots since individual raw concentration data were not available from literature data. Therefore, the model performance was not able to be completely assessed. I added that as limitation in the manuscript.

L242 (Discussion)

The visual predictive check plots confirmed the prediction capability of the model (Figure 1). Therefore, the models are considered to have good predictive performance for PK/PD evaluation.

The visual predictive check plots (Figure 1) indicated that most observed lung concentrations were within model-predicted ranges. However, observed lung concentrations currently available (mean ± standard deviation at three timepoints) were too few to assess the model performance. Furthermore, model validation by goodness-of-fitness plots were not able due to lack of individual raw concentration data in the literature [16]. The model in this study thus needs to be further validated in the future.

Comment 4. Please also add PK for ampicillin and sulbactam, respectively in the Introduction section.

Response 4.

I added about PK for ampicillin and sulbactam in the manuscript.

L36 (Introduction)

The combination of the β-lactam antimicrobial agent ampicillin and β-lactamase inhibitor sulbactam is used at a dose ratio of 2:1. Ampicillin/sulbactam has broad-spectrum antibacterial activity against gram-positive, gram-negative, and anaerobic bacteria, and it has been used as a first-line treatment for pneumonia and preoperative prophylaxis for pneumonectomy [1-4].

Ampicillin/sulbactam, contained β-lactam antimicrobial agent ampicillin and β-lactamase inhibitor sulbactam has been widely used at a dose ratio of 2:1. Both ampicillin and sulbactam are water-soluble drugs and mainly excreted by kidneys [1]. Ampicillin/sulbactam has been used as a first-line treatment for pneumonia and preoperative prophylaxis for pneumonectomy [1-4].

Comment 5. What formula was used for CLcr? CG or MDRD or other?

Response 5.

CLcr was calculated using CG methods. Therefore, the following sentences were added to the revised manuscript.

L364 (Materials and Methods)

Also, CLcr, which was calculated by the Cockcroft-Gault formula, was incorporated as a covariate of CL.

L384 (Footnote of Table 4)

CLcr was calculated by the Cockcroft-Gault formula.

Comment 6. Abbreviations should be used consistently. Use of the "PKPD" abbreviation is intermittent.

Response 6.

I fixed “pharmacokinetics/pharmacodynamics” to “PK/PD” consistently in the manuscript.

L179 (Table 2)

Table 2. Pulmonary pharmacokinetic/pharmacodynamic breakpoints of ampicillin-sulbactam for general treatment with both activities of bactericidal ampicillin and β-lactamase-inhibiting sulbactam

Table 2. Pulmonary PK/PD breakpoints of ampicillin-sulbactam for general treatment with both activities of bactericidal ampicillin and β-lactamase-inhibiting sulbactam

L181 (Table 2)

Note: Pulmonary pharmacokinetic/pharmacodynamic breakpoints are defined as the highest MIC attaining more than 90% probabilities in lung tissue.

Note: Pulmonary PK/PD breakpoints are defined as the highest MIC attaining more than 90% probabilities in lung tissue.

L217 (Table 3)

Table 3. Pulmonary pharmacokinetic/pharmacodynamic breakpoints of ampicillin-sulbactam for A. baumannii treatment with sulbactam activity

Table 3. Pulmonary PK/PD breakpoints of ampicillin-sulbactam for A. baumannii treatment with sulbactam activity

L218 (Table 3)

Note: Pulmonary pharmacokinetic/pharmacodynamic breakpoints are defined as the highest MIC attaining more than 90% probabilities in lung tissue.

Note: Pulmonary PK/PD breakpoints are defined as the highest MIC attaining more than 90% probabilities in lung tissue.

L404 (Materials and Methods)

4.4. Pharmacokinetic/pharmacodynamic simulation

4.4. PK/PD simulation

Comment 7. Why is this model a hybrid? How many compartments are there in this model?

Response 7.

I added the explanation about a hybrid model in the manuscript.

L317 (Materials and Methods)

Hybrid model is a model in which physiological parameters such as organ blood flow and volume are partially connected to the conventional PK model. Based on blood concentrations, this model has been used for the target-site PK/PD analysis [19, 25-28]. In this study, a lung compartment was connected to two-compartment PK model using blood concentrations [19]: thus, the hybrid model consisted of three compartments.

Comment 8. Table 4: Fix effects parameter (X) -> Fixed ~. Several semicolons are missing in the footnote.

Response 8.

I fixed “Fix effects parameter” to “Fixed effects parameter” in Table 4. Also, some semicolons were added in the footnote.

L378 (Table4)

Fix effects parameter

Fixed effects parameter

L379 (Table4)

a, parameters derived from [19] b, parameter derived from lung tissue/serum AUC0-3 h ratio (Table 1) c, parameter derived from [20]

RSE, relative standard error θ, population mean value; η, random variable which is normally distributed with a mean of zero and variance

ε, random error which is normally distributed with a mean of zero and variance

a, parameters derived from [19]; b, parameter derived from lung tissue/serum AUC0-3 h ratio (Table 1); c, parameter derived from [24, 25]

RSE, relative standard error; θ, population mean value; η, random variable which is normally distributed with a mean of zero and variance

ε, random error which is normally distributed with a mean of zero and variance

Comment 9. Various dosing regimen simulations were attempted. Ampicillin/sulbactam is already widely used clinically. What does it mean to subdivide the regimen by MIC and CLcr?.

Response 9.

I added the meaning of individualizing the regimen by MIC and CL in the manuscript.

L56 (Introduction)

According to the pharmacokinetic (PK)/pharmacodynamic (PD) theory, the antibacterial activity of β-lactams such as ampicillin is generally dependent on the exposure time during which the drug concentrations remain above the minimum inhibitory concentration (MIC) for pathogens (T > MIC) [12, 13].

However, ampicillin/sulbactam 12 g/day, as compared with 6 g/day, increased the incidence of hepatobiliary enzyme elevation [12]. Therefore, dose-dependent side effects cannot be excluded, and dosage adjustment considering each patient’s characteristics (e.g. renal function) should be required. Furthermore, according to the pharmacokinetic (PK)/pharmacodynamic (PD) theory, the exposure time which the drug concentrations remain above the minimum inhibitory concentration (MIC) for bacteria (T > MIC) is correlated with the antibacterial activity of β-lactams such as ampicillin [13, 14]. The efficacy should be improved with appropriately adjusted regimens according to the MIC for the bacteria.

Reviewer 2 Report

The authors present interesting conclusions and recommendations especially regarding the use of ampi/tavibactam in A. Bumanni pneumonia, where they recommend doses above the maximum approved dose based on this PK/PD work. However, for me to be able to assess the scientific value of this paper and the evidence underlying this recommendation, the manuscript needs to be improved. Especially concerning the methodology section.

If I understand correctly the authors have tried to synthesize a population PK/PD model from literature data. Reading this paper it is not clear what data was extracted from which papers. What were the demographic characteristics of these studies? It appears that AUC/Cmax values were extracted or somehow calculated (paragraph 4.2) and used to calculate KP values to be incorporated into a PK model (paragraph 4.1)? What do the authors exactly mean with the method “described by Frank et al” (line 383)? For me as reviewer and readers in the future, to be able to properly assess the scientific value of this paper, and what it adds to the current knowledge on ampi/sulbactam pk/pd, the methodology section needs to be improved answering the questions above. Please pay special attention to writing it all down in chronological order.

Some specific questions regarding the model:

The model used in this paper (figure 4): What is the scientific basis for this model? Has it been described elsewhere? This is unclear to me. Why is Qlung used both to describe blood->lung flow and clearance from the lungs? What does this clearance from the lung represent? Isn’t it more physiological to expect an equilibriating Q between Vcentral and Vlung (so vice versa)?  What is the basis of fixing Qlung to 207 L/h? I cannot find this number in the referred paper (ref 20).

The authors claim a good model performance based on the VPC in figure 1 (line 239). I do not necessarily agree. First, there is a very low amount of observations which make it difficult to asses the model performance. Two: how are covariates (for example renal CL) distributed among the observations and does the model indeed capture a covariate effect here? Lastly, most importantly, the observations show little to no clearance from lung tissue over the first 3 hours (one could easily draw a horizontal line through the observations). Therefore, I do not think the authors can conclude that their current model is a good descriptor of the lung PK of ampi/sulbactam, at least not using the current dataset. I suggest the authors show the raw data in figure 1 and/or consider to obtain more data. In the least, they should discuss this limitation in the discussion section.

Some specific questions:

Line 18: “Target attainment”: what is the exact target used?

Line 25: Please define MIC90 in abstract, and in manuscript as well.

Line 398: what covariate distributions were used for the simulations, in how many virtual subjects?

Author Response

Thank you for your comments. The text shown in red below is our response to your thought-provoking comments. Line (L) numbers are shown as those in the revised manuscript. We would be grateful if you could re-review it. Please also refer to the attached file.

If I understand correctly the authors have tried to synthesize a population PK/PD model from literature data. Reading this paper it is not clear what data was extracted from which papers. What were the demographic characteristics of these studies? It appears that AUC/Cmax values were extracted or somehow calculated (paragraph 4.2) and used to calculate KP values to be incorporated into a PK model (paragraph 4.1)? What do the authors exactly mean with the method “described by Frank et al” (line 383)? For me as reviewer and readers in the future, to be able to properly assess the scientific value of this paper, and what it adds to the current knowledge on ampi/sulbactam pk/pd, the methodology section needs to be improved answering the questions above. Please pay special attention to writing it all down in chronological order.

Response.

1. What were the demographic characteristics of these studies?

(Answer)

We added the demographic information of literature data used in this study (Table S1). Therefore, the following sentence and supplementary Table S1 were added to the revised manuscript.

L372 (Materials and Methods)

The demographic information of literature data used in this study was represented in supplementary material (Table S1).

Supplementary material (Table S1)

Please refer attached file.

2. It appears that AUC/Cmax values were extracted or somehow calculated (paragraph 4.2) and used to calculate KP values to be incorporated into a PK model (paragraph 4.1)?

(Answer)

Cmax was defined and extracted as the highest value in mean concentrations described in the literature. AUC0-3h was estimated using mean concentrations described in the literature. The result of the lung tissue/serum AUC0-3 h ratio was used as KPlung. Therefore, the following sentences were fixed to the revised manuscript.

L388 (Materials and Methods)

The lung tissue/serum ratio and ampicillin/sulbactam ratio in serum and lung tissue were estimated as described by Frank et al. [15]. For each drug, Cmax was defined as the observed mean maximum concentration. The area under the drug concentration–time curve from 0 to 3 h (AUC0–3 h) using mean concentrations was estimated according to the trapezoidal rule.

Mean values of lung tissue and serum concentrations (1, 1.5 and 2-4 h each) in patients reported by Frank et al. [16] were used because individual raw data were not described. For each drug, Cmax was defined as the highest value in the mean concentrations described in the literature. AUC0–3 h based on the mean concentrations (three timepoints) was estimated according to the trapezoidal rule.

L371 (Materials and Methods)

KPlung was calculated using the result of the lung tissue/serum ratio as mentioned in Results section.

The result of the lung tissue/serum AUC0-3 h ratio calculated in Results section was used as KPlung (lung-to-plasma partition coefficient).

L90 (Results)

The mean Cmax values for ampicillin and sulbactam were 40.8 and 25.3 μg/mL, respectively, in serum and 35.6 and 8.6 μg/g, respectively, in lung tissue. The mean AUC0–3 h values for ampicillin and sulbactam were 83.5 and 51.2 μg·h/mL, respectively, in serum and 73.6 and 18.9 μg·h/g, respectively, in lung tissue. For ampicillin, the mean lung tissue/serum ratios were 0.873 for Cmax and 0.881 for AUC0–3 h. For sulbactam, the mean lung tissue/serum ratios were 0.339 for Cmax and 0.368 for AUC0–3 h.

The Cmax values for ampicillin and sulbactam from literature were 40.8 and 25.3 μg/mL in serum, and 35.6 and 8.6 μg/g in lung tissue, respectively. The area under the drug concentration-time curve from 0 to 3 h (AUC0–3 h) values for ampicillin and sulbactam calculated from the literature were 83.5 and 51.2 μg·h/mL in serum, and 73.6 and 18.9 μg·h/g in lung tissue, respectively. For ampicillin, the lung tissue/serum ratios were 0.873 for Cmax and 0.881 for AUC0–3 h. For sulbactam, the lung tissue/serum ratios were 0.339 for Cmax and 0.368 for AUC0–3 h.

L127 (Results)

Figure 1. Visual predictive check plots of (A) ampicillin and (B) sulbactam representing the observed lung tissue concentrations after a 15-min infusion of ampicillin/sulbactam 3 g (ampicillin 2 g and sulbactam 1 g) are presented [15]. The heavy line and dotted line denote the median and 95% predicted interval calculated from 1000 replicates, respectively.

Figure 1. Visual predictive check plots of (A) ampicillin and (B) sulbactam representing the observed lung tissue concentrations (mean ± standard deviation) after a 15-min infusion of ampicillin/sulbactam 3 g (ampicillin 2 g and sulbactam 1 g) derived from literature data [16]. The heavy line and dotted line denote the median and 95% predicted interval calculated from 1000 replicates, respectively.

3. What do the authors exactly mean with the method “described by Frank et al” (line 383)?

(Answer)

It meant that we used mean concentrations described in Frank et al. report. Therefore, the following sentence was fixed to the revised manuscript.

L388 (Materials and Methods)

The lung tissue/serum ratio and ampicillin/sulbactam ratio in serum and lung tissue were estimated as described by Frank et al. [15].

Mean values of lung tissue and serum concentrations (1, 1.5 and 2-4 h each) in patients reported by Frank et al. [16] were used because individual raw data were not described.

The model used in this paper (figure 4): What is the scientific basis for this model? Has it been described elsewhere? This is unclear to me. Why is Qlung used both to describe blood->lung flow and clearance from the lungs? What does this clearance from the lung represent? Isn’t it more physiological to expect an equilibriating Q between Vcentral and Vlung (so vice versa)?  What is the basis of fixing Qlung to 207 L/h? I cannot find this number in the referred paper (ref 20).

Response.

1. What is the scientific basis for this model? Has it been described elsewhere?

(Answer)

Hybrid model, in which physiological parameters such as organ blood flow and volume are partially connected to the conventional PK model, has been used in previous studies.

Therefore, the following sentences were added to the revised manuscript.

L316 (Materials and Methods)

The pulmonary pharmacokinetics of ampicillin and sulbactam was separately described for each drug using the following hybrid model (Figure. 4).

The pulmonary pharmacokinetics of ampicillin and sulbactam was separately described for each drug using the following hybrid model (Figure. 4). Hybrid model is a model in which physiological parameters such as organ blood flow and volume are partially connected to the conventional PK model. Based on blood concentrations, this model has been used for the target-site PK/PD analysis [19, 26-29]. In this study, a lung compartment was connected to two-compartment PK model using blood concentrations [19]: thus, the hybrid model consisted of three compartments.

2. Why is Qlung used both to describe blood->lung flow and clearance from the lungs? What does this clearance from the lung represent? Isn’t it more physiological to expect an equilibriating Q between Vcentral and Vlung (so vice versa)?

(Answer)

As you pointed out, we assumed that lung-to-systemic clearance and systemic-to-lung clearance are approximately equivalent to physiological organ blood flow and set as lung blood flow in the hybrid modeling. This is because a specific organ clearance depends mainly on its physiological organ blood flow. Also, we assumed that independently from the conventional PK model, mass balance in the lung compartment was assumed not to affect mass balance in the central and peripheral compartments.

Therefore, the following sentences were added to the revised manuscript.

L238 (Discussion)

Since a specific organ clearance depends mainly on its physiological organ blood flow, both system-to-lung clearance and lung-to-system clearance were assumed to be the same, and set as lung blood flow in the hybrid modeling. Independently from the conventional PK model, mass balance in the lung compartment was assumed not to affect mass balance in the central and peripheral compartments.

3. What is the basis of fixing Qlung to 207 L/h? I cannot find this number in the referred paper (ref 20)

(Answer)

We added the detail of Qlung calculation. Qlung (lung plasma flow) was calculated from lung blood flow of literature data, considering hematocrit value.

Therefore, the following sentences were added to the revised manuscript.

L368 (Materials and Methods)

Since the drug is present only in the plasma portion, lung plasma flow was calculated by multiplying lung blood flow (360 L/h) [24, 25] and human hematocrit value (approximately 42.5%) as follows: Qlung = lung blood flow (360 L/h) * (1-0.425) = 207 (L/h).

The authors claim a good model performance based on the VPC in figure 1 (line 239). I do not necessarily agree. First, there is a very low amount of observations which make it difficult to asses the model performance. Two: how are covariates (for example renal CL) distributed among the observations and does the model indeed capture a covariate effect here? Lastly, most importantly, the observations show little to no clearance from lung tissue over the first 3 hours (one could easily draw a horizontal line through the observations). Therefore, I do not think the authors can conclude that their current model is a good descriptor of the lung PK of ampi/sulbactam, at least not using the current dataset. I suggest the authors show the raw data in figure 1 and/or consider to obtain more data. In the least, they should discuss this limitation in the discussion section.

Response.

As you pointed out, observed lung concentrations currently available were too few to assess the model performance. I added that as this limitation.

Therefore, the following sentences were fixed to the revised manuscript.

L242 (Discussion)

The visual predictive check plots confirmed the prediction capability of the model (Figure 1). Therefore, the models are considered to have good predictive performance for PK/PD evaluation.

The visual predictive check plots (Figure 1) indicated that most observed lung concentrations were within model-predicted ranges. However, observed lung concentrations currently available (mean ± standard deviation at three timepoints) were too few to assess the model performance. Furthermore, model validation by goodness-of-fitness plots were not able due to lack of individual raw concentration data in the literature [16]. The model in this study thus needs to be further validated in the future.

Line 18: “Target attainment”: what is the exact target used?

Response.

We added the exact targets used.

Therefore, the following sentences were added to the revised manuscript.

L17 (Abstract)

Against bacteria other than A. baumannii (general treatment), PK/PD target was set as both 50% time above minimum inhibitory concentration (T > MIC) for ampicillin and 50% T > 0.5MIC for sulbactam. For A. baumannii treatment, PK/PD target was set as 60% T > MIC for sulbactam.

Line 25: Please define MIC90 in abstract, and in manuscript as well.

Response.

I added the definition of MIC90 in abstract and manuscript.

Therefore, the following sentences were added to the revised manuscript.

L26 (abstract)

For treatment other than that of A. baumannii, the pulmonary PK/PD breakpoint for ampicillin/sulbactam 3 g four times daily in typical patients with creatinine clearance (CLcr) of 60 mL/min was 2 μg/mL, which covered the MIC90s of most gram-positive and gram-negative bacteria.

For general treatment, the pulmonary PK/PD breakpoint for ampicillin/sulbactam 3 g four times daily in typical patients with creatinine clearance (CLcr) of 60 mL/min was 2 μg/mL, which covered the MIC90s (MICs that inhibited the growth of 90% of the strains) of most gram-positive and gram-negative bacteria.

L264 (Discussion)

Regarding general treatment with bactericidal activity of ampicillin and the β-lactamase–inhibiting activity of sulbactam, the pulmonary PK/PD breakpoints of the bactericidal target (50% T > MIC for ampicillin and 50% T > 0.5 MIC for sulbactam in lung tissue) for ampicillin/sulbactam 3 g four times daily (12 g/day: approved maximum dosage) were 1 μg/mL (MIC90 of MSSA) for CLcr = 90 mL/min, 2 μg/mL (MIC90 of S. pneumoniae and Prevotella species) for CLcr = 60 mL/min, and 8 μg/mL for CLcr = 30 mL/min.

Regarding general treatment, the pulmonary PK/PD breakpoints of ampicillin/sulbactam 3 g four times daily (12 g/day: approved maximum dosage) were 1 μg/mL (MIC90 [MIC that inhibited the growth of 90% of the strains] of MSSA) for CLcr = 90 mL/min, 2 μg/mL (MIC90 of S. pneumoniae and Prevotella species) for CLcr = 60 mL/min, and 8 μg/mL for CLcr = 30 mL/min.

L424 (Materials and Methods)

These included methicillin-susceptible Staphylococcus aureus (MSSA; n = 676, MIC90 = 1 μg/mL), S. pneumoniae (n = 565, MIC90 = 2 μg/mL), H. influenzae (All strains: n = 544, MIC90 = 4 μg/mL; β-lactamase–nonproducing ampicillin-resistant H. influenzae: n = 70, MIC90 = 8 μg/mL), M. catarrhalis (n = 491, MIC90 = 0.25 μg/mL), S. anginosus group (n = 100, MIC90 = 0.25 μg/mL), Peptostreptococcus species (n = 100, MIC90 = 0.03 μg/mL), Prevotella species (n = 100, MIC90 = 2 μg/mL), Fusobacterium species (n = 50, MIC90 = 0.06 μg/mL) for ampicillin/sulbactam, and A. baumannii (n = 27, MIC90 = 4 μg/mL) for sulbactam.

These included methicillin-susceptible Staphylococcus aureus (MSSA; n = 676, MIC for the 90th percentile of the clinical strains [MIC90] = 1 μg/mL), S. pneumoniae (n = 565, MIC90 = 2 μg/mL), H. influenzae (All strains: n = 544, MIC90 = 4 μg/mL; β-lactamase–nonproducing ampicillin-resistant H. influenzae: n = 70, MIC90 = 8 μg/mL), M. catarrhalis (n = 491, MIC90 = 0.25 μg/mL), S. anginosus group (n = 100, MIC90 = 0.25 μg/mL), Peptostreptococcus species (n = 100, MIC90 = 0.03 μg/mL), Prevotella species (n = 100, MIC90 = 2 μg/mL), Fusobacterium species (n = 50, MIC90 = 0.06 μg/mL) for ampicillin/sulbactam, and A. baumannii (n = 27, MIC90 = 4 μg/mL) for sulbactam.

Line 398: what covariate distributions were used for the simulations, in how many virtual subjects?

Response.

The simulations were performed by producing parameter sets of 1000 virtual subjects each CLcr of patients.

Therefore, the following sentence was fixed to the revised manuscript.

L405 (Materials and Methods)

A set of fixed-effects parameters θi (CL, Vcentral, Q, Vperipheral, KPlung, Qlung, and Vlung) for ampicillin and sulbactam were randomly generated 1000 times using the $SIMULATION command in NONMEM based on each mean value and interindividual variability.

A set of fixed-effects parameters θi (CL, Vcentral, Q, Vperipheral, KPlung, Qlung, Vlung) of 1000 virtual subjects for each renal function (three typical CLcr = 90, 60, 30 mL/min) were randomly generated using the $SIMULATION command in NONMEM based on each mean value and interindividual variability.

Round 2

Reviewer 1 Report

Thank you for accepting my opinion. I have no further comments.

Reviewer 2 Report

Thank you for incoorporating the answers. I still have some reservations regarding the validity of the model for describing lung-tissue concentrations but these limitations are now clearly adressed in the discussion section. I agree with the authors that currently this is the best we can do with the limited data at hand.